# Urban Structure, Subway Systemand Housing Price: Evidence from Beijing and Hangzhou, China

**Xiaoqi Zhang [1], Yanqiao Zheng [1,*], Lei Sun [2] and Qiwen Dai [3,*]**

1   School of Finance, Zhejiang University of Finance and Economics, Hangzhou 310018, China;
    xiaoqizh@buffalo.edu
2   Department Industrial and Systems Engineering, University at Buffalo, Buffalo, NY 14260, USA;
    leisun@buffalo.edu
3   School of Economics & Management, Guangxi Normal University, Guilin 541006, China
*   Correspondence: zhengyanqiao@hotmail.com (Y.Z.); sxsfdx520@163.com (Q.D.)

**Abstract:** Using housing market data of Beijing and Hangzhou, China, we conduct a case study to detect how the difference of urban structure can affect the relationship between the subway system and housing prices. To quantify the characteristics of urban structure, we propose a constrained clustering method, which can not only reveal the spatial heterogeneity of the housing market, but also provides a link between heterogeneity and the underlying urban structure. Applying constrained clustering to Beijing and Hangzhou, we find that the relationship between accessibility to metro stations and housing prices is weak and vulnerable, while the improvement of commuting efficiency, measured by a key variable, the metro index, does have a robust connection to metro premium on housing units. In particular, only a large metro index can be associated with a positive metro premium. Structural features, such as the size of urban core and the existence of multiple sub-centers, influence the metro premium by affecting the value and spatial distribution of the metro index. The evidence from Beijing and Hangzhou supports that in a mono-centric city, the size of the urban core is positively associated with the metro index and the metro premium, while in a poly-centric city with a small urban core, the metro index tends to be lower in the core region and higher in the satellite regions, which enforces the metro premium to be negative in the core while positive outside of the core.

**Keywords:** constrained clustering; hedonic model; housing price; subway system; urban structure

## 1. Introduction

Research on the influence of subways on housing price began in the 1970s when mass construction of rail transport began around the world [1,2]. Most studies demonstrated positive price premiums and a decreasing effect with distance [3,4]. For Chinese cities, some research on Beijing, Shanghai, Shenzhen, and Hong Kong confirm this pattern [5–9]. Despite this, some studies report little or negative effects of subway transit on housing prices, due to noise and crime rates nearby. The work in [10] found that in San Francisco, people relocate away from rail transit to avoid its adverse environmental impacts. The work in [11] found that in Hong Kong, there seems to be no subway effect, because the care for the surrounding environment seems to outweigh the accessibility brought by subway transport.

Although extensive studies have been done to exam the effect of the subway system on the housing price, there still exist some shortages. First, most previous discussions were "context-free" in the sense that they only looked at one single city, especially mega-cities, such as Beijing, Shanghai, and Shenzhen [6,11–13], and the housing data were collected within a neighborhood of one or a couple of subways lines. The "context-free" analysis did not pay enough attention to the complexity

of the relationship between the subway system and the housing market and ignored the potential effect of many important hidden variables, such as the urban structure. Although exceptions exist, such as [14,15], it is typical that only a simple mono-centric city with a prescribed CBD is considered, which is too simple and may not be realistic. Second, except for a few papers [1,14–16], previous studies focused mainly on accessibility to metro stations, which is measured by the distance to metro stations and/or the dummy variables derived from distance, such as whether there is a metro station within the range of one mile [13,17–20]. These studies ignore the heterogeneity among different metro stations and might over-estimate the positive effect of the subway system on housing prices. Even in the case that different commuting times are involved in characterizing the heterogeneity of metro stations [1,14–16], the real increase of commuting efficiency brought by the subway is still not analyzed in a systematic way.

The current study fills these two gaps. This paper compares the relationship between the subway and the housing price in two different cities, Beijing and Hangzhou, China. On that basis, we analyze their urban structure of in a novel way and discuss the connection to the subway system and the housing price. Our study contributes to the existing literature in the following three aspects:

First, this study considers multiple measures of the transportation convenience brought by the subway system. In addition to the accessibility measure to metro stations, a metro index is constructed for every metro station by using the Baidu Direction API and defined as the ratio of duration time by no-subway routes to a set of major destinations in a city versus the duration time by subway-prioritized routes. The use of relative commuting time effectively captures heterogeneity among metro stations in different locations, which cannot be reflected by accessibility only. Moreover, the set of destinations will be selected in a completely data-driven manner on the basis of the spatial distribution of points of interest (POI). Comparing to only including single CBD or a set of employment centers known a priori as the destinations [3,16], the data-driven construction can effectively resolve the issue of subjectivity and better reflect the real traffic demands of local residents.

Second, we thoroughly investigate how hidden features of the urban structure can influence the premium of metro stations on housing units. As urban structure is not naturally a quantitative variable, it is hard to insert into many widely-used analytic tools of housing price, such as the hedonic price model (HPM) and the geographically-weighted regression (GWR). For that issue, a novel constrained clustering technique is proposed to quantify the impact of urban structure, through which we can quantitatively detect which feature of urban structure matters and how it matters. It turns out that the clustering result can be naturally incorporated with HPM. Using HPM together with the metro index and constrained clustering, it is found that features such as the size of the urban core, the existence of multiple centers, and the relative positions between the urban core and satellite centers are all important determinants of the sign and scale of the metro premium on housing price.

Third, we find that there exists strong inter-urban heterogeneity between Beijing and Hangzhou, as well as intra-urban heterogeneity within Hangzhou in terms of the metro premium on housing price. It is an important new finding that the heterogeneity of the metro premium could happen at both inter-urban and intra-urban levels; to the best of our knowledge, it has not yet been documented elsewhere in the literature. This new finding also implies the importance of including both intra-urban and inter-urban housing data simultaneously. In most previous studies, either the city-level aggregated data across multiple cities or intra-city micro data within on single city were used to analyze the housing market [13,21,22]; they are rarely combined together as this paper does.

This paper is organized as follows. Relevant literature is reviewed in Section 2. The study area and data source are briefly discussed in Section 3. In Section 3.3, the detailed construction of the metro index, constrained clustering, and the variables to be used in HPM regression are discussed in detail. Analytic results are reported in Section 4, where the spatial distribution of housing price and subway stations in Beijing and Hangzhou will be discussed. Full sample regression results and sub-sample regression results based on clusters will also be reported and analyzed. Section 5 concludes.

## 2. Literature Review

### 2.1. Subway System, Commuting Efficiency, and Housing Prices

The mechanisms affecting housing prices have attracted considerable scholarly debate from various theoretical perspectives [23,24]. At the micro level, there are three major categories of attributes that are considered as important factors to determine the price of real estate: the structural, neighborhood, and locational characteristics [25–28]. By regressing housing prices against those characteristics in a hedonic model, scholars have identified significant determinants of housing prices [26].

Apart from the attributes of the three categories, it is believed that residential units close to a metro station can enjoy a huge positive premium due to the improvement of commuting efficiency [16,20,29,30]. This is the case especially for cities in China, because traffic congestion widely exists in all major cities of China, which makes access to the rail transit system a valuable resource [15]. Many empirical research works were devoted to investigating the quantitative relationship between accessibility to metro stations and the metro premium on housing units [5,12,13,19,29,31].

As many online platforms, such as http://www.fang.com/, start to provide detailed sales and/or transaction data that are publicly available, a increasing amount of studies has been done on the basis of micro-level housing data in recent years. Both of the cross-sectional and longitudinal data are often used in these studies. Cross-sectional data generate a comparison of housing prices between locations close to and distant from metro stations [13,15,30,32]; longitudinal or penalty data are used to capture the price variation before and after the construction of a metro station in places around that station [2,3,19,33]. In China, dramatic urban development makes housing prices sky-rocket within a short time period [34–36], while the construction of a subway line takes one or two years on average, so longitudinal or panel data cannot effectively distinguish between the effects of urban economic growth and the construction of a subway. In addition, policies change very often in China. Especially in the past decade, policies have alternated several times between restricting and stimulating "bubbles" the housing market of all major cities [37–41], which include those cities with operating subway systems. All the policy changes can significantly impact housing prices, as home-buyers often react strongly to sudden policy changes [36,42]. As a consequence, it is difficult to analyze the metro premium with longitudinal/panel data in China.

In addition to data issues, many studies rely on a hidden assumption that all metro stations are homogeneous in terms of their contribution to commuting efficiency; therefore, very often, only a single proxy of the accessibility to metro stations is considered. However, commuting efficiency is not homogeneous, and it varies significantly along with the location [15], the network structure of the subway system [43], the density of traffic flow, the availability of alternative traffic tools [3,16], and many other factors. More critically, it depends on the choice of destination, and consequently on the commuting demand of local residents. The determinants of the efficiency of metro stations have been widely studied in the literature [43,44], while limited attention has been paid to the connection between commuting efficiency and the premium of a metro station brought to housing units nearby. Therefore, a thorough investigation is required for the relationship among housing price, accessibility to metro stations, and their commuting efficiency.

### 2.2. Spatial Heterogeneity, Urban Structure, and Housing Prices

Spatial heterogeneity, namely non-stationary dynamics across space, refers here to spatial variations in housing prices and household preferences [28,45–47]. The existence of spatial heterogeneity makes the OLS-based hedonic price model (HPM) invalid, because of its stringent assumptions that the coefficients must be constant over all places, as well as its neglect of spatial effects [46,48–50]. Alternatively, local spatial models such as geographically-weighted regression (GWR) have been widely employed to control spatial heterogeneity [28,51].

It is notable that spatial heterogeneity has a close connection to structural features of the urban

area [32,52–54]. For instance, the housing market of a poly-centric metropolis is usually composed of several disequilibrium submarkets [12,24,47,55,56]. Spatial heterogeneity then comes from different determinants of housing prices in various submarkets [30,56–58].

Spatial heterogeneity co-exists with local homogeneity, because a submarket is usually a geographic continuum that corresponds to a local center of the metropolis, and pricing of residential units inside a submarket is homogeneous [24,59]. That means that housing units close to each other geographically are more homogeneous and more likely to stay in the same submarket except on the geographic boundary of different submarkets. Clearly, both spatial heterogeneity and local homogeneity are closely connected to and essentially induced by the poly-centric structure of a city, while their connection cannot be revealed by many popular local spatial models, such as GWR. Because, in the GWR framework, every housing unit is treated as one single submarket, this partition is too fine to keep the necessary homogeneity within neighborhoods; thus, it cannot reflect the connection between the hidden features of urban structure and housing market. Alternatively, delineating the housing market into different areas and estimating separate hedonic equations for each submarket constitute a useful methodology; it can better balance the spatial heterogeneity and local homogeneity [55,60] and reveal the hidden influence of urban structure. However, figuring out a division of submarkets that can make sense both geographically and economically is not trivial; many empirical studies have directly applied partitions based on prior knowledge, such as the zip-code zones and administrative districts [56,57]. Such a choice is somewhat subjective, and so, its validity is not properly verified. A more data-oriented partition method is required to resolve the issue of subjectivity. Many works have been done in this direction [47,54], but none of them were designed for the metro premium on housing prices; thus, further explorations are needed.

## 3. Study Area, Data, and Methodology

### 3.1. Study Areas

This paper considers Beijing and Hangzhou in this case study, because they are distributed in different parts of China and relatively independent of each other both geographically and economically. They are the greatest or one of the greatest local centers and have different development stages and different urban structures. In addition, the history, coverage, and distribution density of the subway system in Beijing and Hangzhou are quite different. Therefore, we believe comparing these two cities could yield a more versatile picture of the relationship between the subway system and housing price.

As the capital of China, Beijing is enclosed by ring roads. The officially declared CBD is the *Guomao* center, lying between the southeast second- and third-ring roads. In addition to CBD, there are multiple commercial centers in Beijing with an extremely high-dense population, such as Zhongguan Village in Haidian district, which is also known as the "Silicon Valley" of China, as well as where the most well-known Chinese universities, Peking University and Tsinghua University, are located. There are several satellite centers lying around the suburb or exurb of Beijing, such as the centers in Changpin district in the north and in Tongzhou district in the southeast.

Hangzhou is the capital of Zhejiang Province, China. Its major urban area is around the east coast of the well-known West Lake, and its city core is centered at Wulin Square. There is a new CBD (Qianjiang New City) under construction, which is located next to the southeast boundary of the city core and sits on both sides of Qiantang River. In the east of city core are three satellite sub-cities, Linping in the north, Xiasha in the middle, and Xiaoshan in the south. Among them, Xiasha was an industrial zone and is planned to become a college park and a new economic development zone of Hangzhou; Xiaoshan is next to Qianjiang New City, and it is most likely, among the three satellite centers, to experience a fast development in the near future due to urban expansion.

The other background information regarding the economies, and population, subway constructions of Beijing and Hangzhou are summarized in Table A1, and more detailed information is

available in Chinese from the China city statistical yearbook [61] and the various city development reports online.

A visualization of the study areas, Beijing and Hangzhou, is also provided by the map of Figure A1, in which the spatial distribution of sampled housing units is included as well. From Figure A1, it is clear that the spatial distribution of apartments in our dataset roughly reflects the development status of different regions inside the two cities. For Beijing, the main portion of housing units is located within the area enclosed by the sixth-ring road or in neighborhoods along with subway lines. These regions are also the most well-developed parts of the city. In addition, if only looking at the region within the fifth-ring road, it is obvious that the distribution of housing units is quite even, which coincides with the fact that all places in this region are almost evenly developed. In contrast to Beijing, the distribution of housing units in Hangzhou is less balanced. There are clearly three or four centers with a relatively high density of apartments. The core region of Hangzhou is also the place that apartments are distributed most densely. The second dense area is on the southeast coast of Qiantang River, where Qianjiang New City is located. The last center lies on the east and northeast part of the city core where the two satellite sub-cities, Xiasha and Linping, are located.

### 3.2. Data

The dataset for Beijing and Hangzhou was collected from http://www.fang.com/. As is known, http://www.fang.com/ is the largest and most famous online information platform that provides detailed sales and transaction information of new and second-hand apartments in most cities of China. The price and availability information on http://www.fang.com/ is updated in a timely manner; therefore, it catches up with the market dynamics very well. For the purpose of this study, we only collected a cross-sectional dataset recording all the information of on-sale second-hand apartments listed on http://www.fang.com/ during the week ending 29 October 2017. This study focuses on second-hand apartments, because the newly-built apartments are mainly located in suburbs, which potentially leads to a biased result. From each listing, its price, area, age, orientation direction, total floor, floor proportion, and address are extracted, which constitute the set of constructional variables that will be used in HPM regression.

As discussed in Section 2.2, covariate variables involved in the regression analysis are divided into four categories, which cover the constructional, neighborhood, locational characteristics, and the variables related to commuting. Except for constructional characteristics, variables in the other three categories are not directly available from http://www.fang.com/; they have to be constructed from the geographic locations of metro stations, bus stations, schools, tertiary hospitals, large shopping malls, and major destinations of a city, which are all collected with Baidu Place API (a full description of the API and the other Baidu APIs used in the study can be found at the url: http://lbsyun.baidu.com/). The detailed definition and construction of variables used in HPM regression are left to Section 3.3.3.

In addition, the accurate geographic location of every listed apartment is also needed, while http://www.fang.com/ only provides a description of their addresses. Therefore, Baidu Geocoding API was used to extract the longitude and latitude information from the address. After removing those apartments with inaccurate geographic location and/or having an abnormal high price, there were 2359 and 4130 apartments remaining for Beijing and Hangzhou respectively, which comprise the sample that will be analyzed. Their spatial distribution has been sketched in Figure A1.

### 3.3. Methodology

#### 3.3.1. Hot-Spot Analysis and Metro Index

The hot-spot analysis tool is used to calculate the Getis–Ord $Gi^*$ statistic for each feature within a dataset. The resultant z-scores and *p*-values suggest where spatial clusters of features with either high or low values can be found. To be considered as a statistically-significant hot-spot, a feature has to have a high value and simultaneously be surrounded by other features with high values. A higher

z-score corresponds to more intense clustering of high values (a hot-spot). Conversely, a lower z-score corresponds to more intense clustering of low values (a cold spot). The Getis–Ord local statistic is given as:

$$G_i^* = \frac{\langle w_i - \overline{w_i}, x - \overline{x} \rangle}{S_x \cdot S_{w_i}} \tag{1}$$

where $\overline{x}$ ($\overline{w_i}$) is the empirical mean of the vector $\{x_1, \ldots, x_n\}$ ($\{w_{i1}, \ldots, w_{in}\}$). $x$ ($:= \{x_1, \ldots, x_n\}$) is the vector representing a feature associated with every location $j \in \{1, \ldots, n\}$; $w_i$ ($:= \{w_{i1}, \ldots, w_{in}\}$) is the weight vector associated with a location $i$ with every $w_{ij}$ being the weight assigned by location $i$ to location $j$. $S_x$ ($S_{w_i}$) are the empirical standard deviations associated with the vector $x$ and $w_i$, respectively. $\langle ., . \rangle$ denotes the inner product of two vectors in the $n$-dimensional Euclidean space.

In our setting, the feature $x$ is chosen to be the empirical distributional density of points of interest (POI), where the POI data are collected from Baidu place API and the types of POIs are selected to include those places where people are heading, such as shopping malls, hospitals, schools, restaurants, and so on. The detailed construction of the spatial density of POI and Getis–Ord statistic in this paper is as below:

First, because of the discrete distribution of points [62,63], grids are used to replace points to analyze the spatial patterns of POIs [64]. The study area is divided into a set of 1 km × 1 km square sub-regions, and then the grid is applied (preliminary analysis shows that 1 km is the optimal choice because a grid size larger than 1 km tends to mask local differences and because a grid size smaller than 1 km can exaggerate local characteristics).

Second, the Gaussian kernel density method is applied to estimate the empirical density of POIs at every grid, and the density is used as the feature value $x_j$ for the $j$th grid. Formally:

$$x_j = \frac{1}{mb^2} \sum_{i=1}^m K\left(\frac{(cp_i - c_j)}{b}\right) \tag{2}$$

where $cp_i$ is the geographic coordinate, (latitude, longitude), associated with the $i$th POI and $c_j$ is the geographic coordinate associated with the $j$th grid. $m$ is the total number of all POIs in the collection. $K$ is the standard two-dimensional Gaussian density function with zero mean and the covariance matrix being the identity matrix. $b$ is the kernel width, which is selected to be $m^{-\frac{1}{3}}$ because such a choice guarantees that as $m \to \infty$, the empirical density converges to its true value in probability.

Finally, the weight $w_{ij}$ is also selected from the Gaussian kernel functions as below:

$$w_{ij} = \frac{1}{b'^2} \cdot K\left(\frac{(c_i - c_j)}{b'}\right) \tag{3}$$

with the kernel width $b'$ selected as $m'^{-\frac{1}{3}}$ where $m'$ is the total number of grids. The reasoning for the choice of $b'$ is the same as that of $b$ for the empirical density (2).

The resulting Getis–Ord statistic ($G_i^*$) reflects the spatial pattern of the clustering intensity of POIs within a city. The grids with a significantly positive value of the Getis–Ord statistic are associated with the places where residents more frequently go and therefore attract the most traffic flow. Applying the K-means clustering to that set of grids, $K$ different positions are extracted and can be interpreted as the major destinations of the entire city. In the current paper, we set $K = 18$ because based on a preliminary experiment, 18 is the greatest number that can guarantee all destinations not too close to each other in both Beijing and Hangzhou. The spatial distribution of 18 major destinations is plotted in Figure A2, Section 4.1.

Both locational characteristics of housing units and the metro index are constructed from the 18 major destinations of each city. As will be shown in Table A2, Section 3.3.3, five variables, duration PCA$i$ for $i = 1, \ldots, 5$, are used to represent the locational characteristics of housing units; they are simply the greatest 5 principal components of the logarithm of commuting time taken from a given

housing unit to all of 18 destinations. The key variable, the metro index, for every housing unit is defined to be the metro index of its nearest metro station, while for a metro station, the metro index is defined by the following equation:

$$metro\ index_i = \frac{1}{18} \sum_{j=1}^{18} \frac{route1_{ij}}{route2_{ij}} \tag{4}$$

where $route1_{ij}$, $route2_{ij}$ are the commuting times taken from station $i$ to destination $j$ by the optimal no-subway route and optimal subway-prioritized route, respectively. The choice of optimal route and the corresponding commuting time are calculated using the Baidu direction API. From the definition (4), it is clear that the greater the metro index is, the higher commuting efficiency a metro station can induce with respect to the residential units nearby.

We believe the metro index constructed in (4) outperforms many alternative indices in the literature in two aspects: first, it provides a more accurate measure of the commuting efficiency attributed to the subway because it compares the difference in time between the subway and alternative traffic tools; in contrast, the previous studies focused exclusively on the commuting time by subway only [15,22]; second, the choice of destinations of traffic routes is completely data-oriented, rather than as many relevant studies have done; CBD and/or commercial centers are selected as the unique destinations [3,15]. The latter method is subjective and cannot fully capture the commuting demand within a city.

### 3.3.2. Hedonic Model

The hedonic price model (HPM), based on ordinary least squares, has been widely used in previous studies on housing prices [25,65]. HPM has the following form:

$$log\ (P_i) = \beta_0 + \beta_1 \cdot Z_i + \beta_2 \cdot X_i + \mu_i \tag{5}$$

$P_i$ refers to the housing price per square meter of residential unit $i$. $Z_i$ includes unit $i'$s construction characteristics, locational characteristics and neighborhood characteristics. $X_i$ includes the commuting-related variables, including accessibility variables to the metro, bus variables, and the metro index. Without loss of generality, all the continuous variables will be rescaled by taking the logarithm, because under the log scale, every regression coefficient can be interpreted as the ratio of the percent change of the corresponding variable versus the percent change of housing price. Throughout the remaining sections, it always refers to its value in the log scale whenever a continuous variable is mentioned, unless there are extra explanations.

### 3.3.3. Variables

The variables in the constructional, neighborhood, locational, and commuting categories are provided in this section. The construction category contains six variables, such as total area, building age, the number of living rooms and bed rooms, the orientation direction, and the floor.

In the neighborhood category, there are three variables that measure the linear distance from a given residential unit to the nearest shopping center, hospital, and schools, where only primary and elementary middle schools are counted as they are the most relevant to housing prices in China [66].

In the location category, this study mainly is concerned with the commuting time from a given place to 18 major destinations of a city with the destinations being constructed as in Section 3.3.1. To eliminate the potential multicollinearity induced by spatial correlation among different destinations, principal component decomposition (PCA) is applied to the logarithm of all the commuting time variables and keeps only the greatest five principal vectors, from which five duration PCA variables are constructed. It turns out that the greatest five components can cover more than 95% of the total variance of all commuting time variables, so they have encoded sufficient information.

Variables in the commuting category include distance to the nearest metro stations, bus stops, and the total number of bus lines at the nearest bus stops. In addition, a set of distance dummies is constructed from the distance variables to reflect the effects of living close to transportation hubs. In addition to standard distance variables, a key variable, the metro index, is inserted into the commuting category, which is calculated and included to capture the spatial heterogeneity of commuting efficiency induced by different metro stations. The detailed construction of the metro index has been given in Equation (4).

The formal definition of relevant variables and their summary statistics is described in Table A2.

### 3.3.4. Constrained K-Means Clustering

As discussed in Section 2.2, HPM neglects the potential spatial heterogeneity within an entire city, while its alternative, GWR, fails to reflect the local homogeneity within a submarket. In this section, we develop a novel method lying between HPM and GWR. The new method combines HPM and the standard K-means clustering by adding a set of constraint conditions to the optimization problem associated with K-means clustering. The constraint conditions depend on the features of all elements within a cluster. The new method is called constrained clustering; it can identify the homogeneity within local submarkets, if there are any, by fully using the information encoded in housing prices and the other covariate variables. Meanwhile, it reveals the spatial heterogeneity of the global housing market in a city. In addition, the resulting clusters turn out to bridge the structural features of a city and its housing market.

The constrained clustering partitions the urban area of a city into a set of sub-regions such that every sub-region can be identified with a housing submarket. It is based on applying the standard K-means clustering to the dataset of sampled housing units. Therefore, the resulting clusters are essentially the clusters of housing units instead of the real "regions" on the map, while the geographic range of every cluster is identifiable as a region if the similarity function of K-means clustering has taken the form of the Euclidean distance on the map. However, using Euclidean distance might lose important economic information of the housing market, because in that distance function, only the geographic coordinates of every housing units are involved, and all the other attributes are irrelevant. Consequently, the resulting clusters can only help identify the geographic homogeneity within neighborhoods, which does not necessarily agree with the economic homogeneity, as we want. To add economic meaning to every cluster, a set of constraint conditions can be incorporated into the standard K-means such that every resulting cluster has to satisfy them. The desired economic homogeneity is then carried out by the newly-added constraints and can be reflected by the shape of the resulting clusters. Formally, a constrained clustering is defined through a constrained optimization problem as below:

$$\min_{K, \mathcal{S}_K} \sum_{S \in \mathcal{S}_K} \sum_{x \in S} (x_c - \overline{x}_{c,S})^2 \tag{6}$$

$$s.t. \begin{cases} f_j(x_S) > 0, \, j = 1, \dots, m \\ S \in \mathcal{S}_K, K \geq 1 \end{cases} \tag{7}$$

where $\mathcal{S}_K$ is a $K$-fold partition of the entire sample and $x$ is the feature vector representing the value of all features associated with a housing unit in the sample. The features should include all variables listed in Table A2 and the 2D geographic coordinates of the unit. The dimensions associated with geographic coordinates are denoted as $c$, and $x_c$ represents the projection of the vector $x$ onto the $c$ dimensions. Note that under these notations, (6) is exactly the objective function for the standard K-means clustering with the similarity function taken to be the Euclidean distance on the map.

In addition, $x_S$ in (7) is the matrix with each column being the single feature vector associated with all housing units in cluster $S$. $f_j$ for $j = 1 \dots, m$ are functions that form $m$ constraints for every cluster $S$. Notice that $f_j$ depends on the features of all housing units within a cluster; in other words, the constraints are "global" at the cluster level, although they are still local at the city level. The local

"globalness" is designed to capture the fact that homogeneity should only happen within a submarket, meanwhile it should be contributed to by all elements within the submarket rather than depending only on some of them.

To be general, let the minimization in (6) be taken over all positive $K$ and $K$-fold partitions because usually, it is not known a priori how many submarkets should be there for a given city. The constrained K-means clustering is implementable through a heuristic algorithm similar to the classic Lloyd algorithm, and the detailed coding for this algorithm can be found in the Python code Supplementary Materials to this paper.

Three points warrant notice: First, constrained clustering consists of two parts, "clustering" and "constraints", which are linked to the spatial heterogeneity and local homogeneity of a housing market, respectively. In the "clustering" part, the existence and extent of spatial heterogeneity are naturally associated with the existence and number of different clusters, while the shapes and relative locations of these clusters give hints about the sources of spatial heterogeneity and their connection to deep-level structural features of a city. In the "constraints" part, the constraint conditions filter elements that should be assigned to a cluster from those that should not; therefore, this reveals what "homogeneity" stands for within a cluster.

Second, different constraints are associated with different types of local homogeneity and accordingly different types of spatial heterogeneity. For instance, consider the constraint by taking $f_1(x_S) = a - var(x_{p,S})$ with $m = 1$, $a > 0$, and $x_{p,S}$ being the column vector associated with housing prices in cluster $S$, then the homogeneity corresponding to that constraint is just the homogeneity in terms of price fluctuation. In contrast, taking $f_j(x_S) = a - var(x_{d_j,S})$ with $m = 5$, $x_{d_j,S}$ being the column vector associated with variable duration PCA$j$ in cluster $S$ reflects the homogeneity in terms of the location relative to major destinations of a city. It is possible to take more complicated functional forms for the constraints, as we will do in Section 4.2, then much deeper-level structural features of a city can be uncovered.

Finally, HPM can be naturally combined with constrained clustering through regressing housing price with respect to the covariates within every cluster. Then, there would be one set of regression coefficients associated with one cluster, while coefficients for different clusters are different in general, which reflects the distinction of the pricing mechanism within different submarkets. Analysis of the variation of regression coefficients among clusters can shed light onto the spatial heterogeneity and its connection to urban structure.

## 4. Result

In this section, the spatial analysis result regarding the spatial distributional patterns of housing units, their prices, and metro stations will firstly be shown. Then, HPM is estimated based on the full sample of Beijing and Hangzhou. As will be shown, Beijing has all key regression coefficients behaving normally as expected, while none of the metro-related coefficients of Hangzhou lie in their expected ranges, which is an indicator of the existence of spatial heterogeneity. Finally, a particular constraint condition is introduced that is based on the metro index and applies the constrained clustering to the sample of Hangzhou. The estimated coefficients are reported by clusters, and the implications of the clustering and regression results are discussed.

### 4.1. Spatial Distribution of Housing Price

Figure A2 displays the spatial distribution of subway stations, unit price (yuan/m$^2$), and the set of major destinations in Beijing and Hangzhou.

Compared to Hangzhou, the subway system in Beijing is much more developed and has better coverage, so that almost all places in the core region of Beijing have convenient access to the subway system. In contrast, in Hangzhou, up to the date of data collection, there were still large regions lacking access to the subway.

The spatial distribution of housing prices reflected by the grayscale in Figure A2 agrees with the distributional intensity of sampled housing units in Figure A1. In Beijing, the mean unit price takes its highest value within the fifth-ring road area and decreases outward. The region with the highest prices lies on the northwest corner of the city core (across the forth-ring road), where Zhongguan village is located. Zhongguan village is famous as the "Silicon Valley" of China, around which are located the most renowned universities, Peking and Tsinghua University, and the best primary and middle school districts in Beijing. The high price in this region reflects the high evaluation of the valuable educational and industrial resources by housing markets.

In Hangzhou, sampled housing units are distributed most densely in its core region (the east coast of West Lake), which also has a relatively high price on average (compare Figure A1 with Figure A2). Therefore, the spatial intensities of the housing market and price basically agree with each other, while the highest price does not appear in the core, the "red color" region in Figure A2 lies on the northwest coast of Qiantang River, which is away from the city core, but coincides with Qianjiang New City, which is planned to be the new CBD of Hangzhou and is under construction currently. The high price appearing in this region somewhat reflects the market expectation.

## 4.2. Full-Sample Regression

Table A3 shows the estimation results of the full-sample HPM, where we focus our attention on the metro-related variables that have been highlighted with bold font in Table A3.

From Table A3, the signs of accessibility variables to metro stations (metro lt1 and metro lt 2) are positive for Beijing and imply a positive premium by living close to metro stations on housing prices. The positive sign agrees with the results reported in previous empirical studies [5,13,31], and is consistent with the positive sign of the metro index and the theoretical point of view that the positive premium comes from the improvement of transportation convenience.

In contrast to Beijing, Hangzhou shows a completely opposite situation: the coefficients for metro lt 1 and metro lt 2 are significantly negative, which conflicts both the positive effect hypothesis and the "bell"-shape effect [13] hypothesis on the relation between accessibility to subway and housing prices. If one looks at the metro index as well, this becomes more controversial. The coefficient of the metro index is negative in Hangzhou, which seems to support a ridiculous relation that with all others being equal, the greater commuting time saving induces a lower price for residential units nearby. Such a negative feedback contradicts the theoretical forecasts in most classic literature in the field of urban economics [67–69].

Interpretation

One explanation of the weird relation between accessibility to the subway, the metro index, and housing prices is the existence of spatial heterogeneity. More precisely, there might exist multiple housing submarkets in Hangzhou. The pricing mechanisms within different submarkets are different, and putting data from different submarkets into one single HPM pollutes the correct relation between housing prices and its determinants.

As residential units in a submarket are geographically close to each other [59], the pricing mechanism in them is homogeneous [56,57]; the constrained clustering perfectly fits into the task of detecting the number and range of housing submarkets.

A specific constraint condition is needed for the implementation of constrained clustering for Hangzhou. A reasonable choice of constraints is to require that the coefficient of the metro index must be positive within every cluster. This is because most literature in urban economics [67–70] supports such a principle, that with all others being equal, the increase in commuting efficiency, or equivalently, the saving of commuting time, should have a positive effect on housing prices. Applying that principle yields a null hypothesis that "the violation to the positive effect principle is a result of spatial heterogeneity and partition of submarkets should eliminate spatial heterogeneity". Then, by the logic of the hypothesis test, a sufficient criterion to verify that null hypothesis becomes:

the positive effect must be observable within every correctly-identified submarket, or equivalently, the estimated coefficient of the metro index must be significantly positive within every cluster. Such a criterion can be written formally as a set of mathematical constraints as below:

$$
\begin{cases}
\hat{\beta}_{S,m} > 0 \\
P_{\beta_{S,m}} < \alpha \\
\det\left(x_{-1,S}^{\top} x_{-1,S}\right) \neq 0 \\
S \in \mathcal{S}_K \\
K \geq 1
\end{cases}
\tag{8}
$$

where $\hat{\beta}_{S,m}$ represents the estimated coefficients of the metro index for a cluster $S$, $P_{\beta_{S,m}}$ is the *P*-value associated with that coefficient, and $\alpha$ denotes the significance level, which will be taken as 0.1; we did not select a finer significance level because all that is needed is just to exclude the negative premium induced by the metro index, and it is totally admissible that there is no significant connection between the subway system and housing prices given that a housing unit is far away from subways. For this purpose, it is not necessary to impose a high significance level on the positivity.

$S$, $\mathcal{S}_K$, and $K$ in (8) have the same meaning as in (7). det denotes the determinant of a matrix; $x_{-1,S}$ is the feature matrix corresponding to cluster $S$, whose column vectors are identified with the variables in Table A2 with the first column, price, being removed. The third line in Constraint (8) is a technical requirement on the size of every cluster, which should not be too small to guarantee the existence of OLS estimators. The form of this condition comes from the mathematical expression of OLS estimators [71].

### 4.3. Constrained Clustering and Urban Structure

#### 4.3.1. Regression by Clusters

Applying the constrained K-means clustering to Hangzhou yields the result shown in Figure A3, where Hangzhou is divided into two sub-regions. One of them covers the core region of Hangzhou; therefore, it is named the "core" cluster; the other basically ranges over three satellite sub-cities, Xiasha, Xiaoshan, and Linping, so it is called the satellite cluster. The clustering result more or less confirms our observations from Figure A1. For the robustness check, the algorithm is re-run 20 times with random initialization; the number of clusters returned is identical, and the range of every cluster is close for all 20 runs. These results support the robustness of clusters plotted in Figure A3.

Corresponding to every cluster, an HPM is estimated; thus the coefficients for different clusters would be different, as suggested in [56], in order to reflect the market segmentation. The full results are reported in Table A4, where the estimated values for subway-related coefficients are highlighted with bold font.

Compared to Table A3, the goodness of fit was improved (measured by Adj. $R^2$), while the signs of the metro index were back to positive and 10% significant for both clusters as required by the constraint (8), which reflects the positive relation between commuting time saving and land value [70].

An interesting finding from Table A4 is that the two clusters demonstrate two controversial types of relation between accessibility to the subway and housing prices. The satellite cluster has had a housing price response that is "normal" and positive for high accessibility to metro stations (the coefficient of metro lt 2 is positive), while the core cluster in Hangzhou still has a negative response despite the positive relation between price and the metro index. Although this negative relation observed in the core cluster is explainable by the argument that living close to a metro station is accompanied by a worse living environment, which lowers the utility increment brought by transportation convenience [17], the co-existence of positive and negative metro premiums within one city does need further exploration.

In the literature, one widely-accepted explanation of the variation of the metro premium within a city is the center-suburb development gap and the spatial variational demand for the subway [15], because low economic development and a high evaluation of the accessibility to the metro are commonly observed for suburban areas, while the opposite is more often observed in urban centers. Such an explanation applies to Hangzhou as well. Xiasha and Linping are two new sub-cities in Hangzhou, whose development level is far behind the urban core; consequently, the accessibility to metro stations has a more supportive impact in these two regions on housing price than in the core.

Although the imbalanced development inside Hangzhou contributes as a crucial factor to the relation between the subway system and housing price, it seems insufficient to explain the significant negativity of the metro premium in the core cluster. Because in Beijing, the unique cluster covers its entire core region (and its boundary), unlike the core cluster in Hangzhou, the metro premium in Beijing is significantly positive (measured by the positive coefficient of metro lt 1 in Table A3). There is a divergence of the metro premium between the cores of Beijing and Hangzhou. Apparently, this divergence is irrespective of the division between city core and suburb, so it cannot be explained by the center-suburb gap. To understand the divergence, a further investigation of urban structure is needed apart from the imbalanced center-suburb relation.

Based on the comparison of the clustering results, the plot of the spatial distribution of the metro index in Hangzhou and Beijing (Figure A4), and the distinction in terms of the development stage of the two cities as summarized in Table A1, we find that:

(1) It is the scale of the metro index rather than the gap between city core and satellite sub-cities that determines the sign of the metro premium: namely, the clusters with a relatively large metro index are more likely to have a positive metro premium, while for regions where the metro index is close to one, the metro premium is more likely to be negative. In fact, Finding (1) is evident from Figures A3 and A4. In addition to having positive metro premiums, both the core cluster of Beijing and the satellite cluster of Hangzhou share one thing in common: their average metro index values are significantly greater than one, meaning that metro stations in these two regions can significantly increase the commuting efficiency for residents nearby, therefore generating great positive externalities. In contrary, the metro index level in the core cluster of Hangzhou is low and close to one, meaning that taking the subway is almost indifferent to ground traffic in terms of commuting to major destinations. As a consequence, the overall effect of living close to a metro station is negative once the negative impact induced by metro stations on the living environment is taken account.

(2) The metro index presents a channel through which a set of more fundamental features of urban structure implement their impacts on housing prices. Among those hidden features, the size of the core region of a city and the existence of satellite sub-cities in the suburban area are exceptionally influential. It is clear from Table A1 that there exists a huge gap between Beijing and Hangzhou in terms of the size of city cores measured either by their areas or the population scale. Namely, Beijing has a core region triple the size of that in Hangzhou, while the population of Beijing is about 2.5-times that of Hangzhou. The spacious core region of Beijing enlarges the traffic demand from place to place inside the core, which makes the underground subway system attractive because of its advantage in speed compared to alternative ground transportation. The dense population strengthens this advantage because a greater population size is always associated with more severe traffic congestion, which further lowers the commuting efficiency of the ground traffics and makes the total commuting time uncertain, which is detrimental for traffic demands for business purposes. In contrast, the city core of Hangzhou is very tiny; it is centered on Wulin Square and extends outward. Therefore, due to the existence of West Lake in the west, Qiantang River in the south and east, and a mountain area in the northeast, there exist natural boundaries for the expansion of the core region of Hangzhou. As a consequence, the urban core of Hangzhou is bound to be small, and it allows only a few metro stations to reside in the tiny core region. Then, the accessibility to metro stations inside the core is not as good as in Beijing. More importantly,

the limitation of space restricts the speed advantage of the subway in contrast to the ground traffic and reduces the traffic demand to the underground subway system, because most of the major destinations inside the core are not distant (see Figure A2b). In that case, the ground transportation, such as bus and taxi, are not that inefficient if the time wasted during the entry and exit of metro stations is considered. Especially for those destinations within the range of riding a bicycle and/or walking, taking the subway might be more time-consuming. In summary, the size of the city core is influential on the comparative advantages of the subway system and its demand, which further determines the level of the metro index and the metro premium with respect to housing units.

The existence of satellite sub-cities is also critical. The co-existence of a core region and satellite sub-cities is more likely to cause the segregation between the residential zone and working zone, which is verified by the instance of Hangzhou, as most of the major destinations in Figure A2b are agglomerated in or next to the core region; they are distant from the three satellite sub-cities, Xiashang, Linping, and the southeast part of Xiaoshan, represented by the satellite cluster in Figure A3. Separation between destinations and satellite sub-cities makes rapid transportation demanding within sub-cities, and the underground subway system only provides the fastest option for local residents. As a consequence, the metro index and metro premium are significantly high in the region covered by the satellite cluster.

In sum, both the size of city cores and the co-existence of a city core and satellite sub-cities are influential on the metro index and metro premium by affecting the relative advantages of multiple traffic tools and the traffic demand within different parts of a city.

### 4.3.2. Implication for Urban Structure

If were revisit Constraint (8), it is clear that the coefficient of the metro index not only depends on the value of the metro index, but also on all the other covariate variables, because the OLS estimator for every single coefficient is essentially a function of the full input data matrix. Therefore, apart from the housing market, many important pieces of information regarding urban structure are also encoded by this constraint, such as the relative position between the living zone and CBD, education centers, industrial centers, and so on. Consequently, the resulting clusters should be able to reflect many more hidden structural features of a city than there appear to be. The trend of integration and segregation among different sub-regions within a city is one such hidden feature. Hangzhou is a good example of this.

An interesting observation for Hangzhou is that: the boundary between the satellite and core cluster on the coast of Qiantang River (covered by the shadow in Figure A3) is artificial somehow, because it does not agree with the river, which is always considered as a natural geographic boundary. Qianjiang New City, which is planned to be the new CBD and located in the northwest side of Qiantang River, is assigned to the satellite cluster and segregated from the core cluster. This fact somehow reflects the fact that despite being the new CBD of Hangzhou, Qianjiang New City is still under construction, and its current development is far behind the old center, represented by Wulin Square, in terms of residential density and commercial activities. All this information has been encoded into housing prices and their determinants, which make the pricing mechanism around the new CBD more analogous to that in satellite sub-cities instead of the urban core. Consequently, due to market forces, Qianjiang New City is segregated out of the core region.

In sum, the observation of Hangzhou shows that the boundary of clusters resulting from the constrained clustering operation can encode the integration/segregation trend of a region that is driven by economic and/or market forces, which cannot be captured if only concerned with geographic and/or administrative boundaries.

*4.4. Policy Implications*

Results from the analysis in previous sections and our analytic methodology have very useful implications for both urban planners and professional real estate appraisers.

For urban planners, promoting the local housing market is usually a major target, and building a subway system is thought to be one efficient option [72,73]. However, our findings display a contrary picture. In particular, it implies that building a subway system may even worsen the situation for a wide range of Tier-2 and/or Tier-3 cities in the middle and west of China, because most of these cities share a common property to Hangzhou, where the city core is tiny and its expansion is strictly restricted by mountains, rivers, and the like. In a city of this kind, a subway system may not be a remedy to traffic issues; to the contrary, it may "pollute" the local living environment and reduce the attractiveness of its housing market.

In addition, our findings imply that to get a positive feedback between the subway system and housing market nearby, distributional patterns of metro stations have to be delicately designed, which has to take into account local traffic demand fully, on the one hand, while guaranteeing relatively fast access to major destinations of a city, on the other. However, these two goals sometimes contradict each other and by and large are affected by the urban structure of every specific city; thus, planning of a subway system has to take full advantage of city-specific features, which has not yet attracted enough attention in the policy-related and urban-planning literature.

For professional real estate appraisers, the clusters resulting from the constrained clustering technique provide valuable information regarding the spatial variation of the price mechanism driven by market forces. This information is less affected by subjective judgments, and thus, it can lead to a more accurate price structure, by which real estate appraisers can better guide rational investment that comes from home-buyer's true residential preference and demand instead of speculation motivation.

## 5. Conclusions

This paper enhances the knowledge and information of studying the total influence of a subway system on housing price in the following aspects:

(1) This study adds a new variable, the metro index, as a measure of the improvement of transportation convenience brought by the existence of a subway system. By including this new variable, the spatial heterogeneity of metro stations can be incorporated into the analysis, which complements the literature that focuses solely on the accessibility measure to metro stations and neglects their spatial heterogeneity. In addition, the metro index is constructed in a completely data-driven manner, which helps avoid subjectivity in selecting destinations.

(2) Urban structure is taken into account as an important hidden variable that may significantly affect the relation between a subway system and housing prices. In order to reflect the variation of urban structures, a case study is conducted of two major cities in China, Beijing and Hangzhou. Furthermore, to quantify the influential features of the urban structure, a constrained clustering technique is proposed; it utilizes the OLS regression coefficient of the metro index as a constraint and adds it to the standard K-means clustering. It turns out that the resulting clusters can reflect the deep-level properties of urban structure, say the integration trend of a region. To the best of our knowledge, this paper is the first attempt to apply constrained clustering to detecting urban structure.

(3) The method, constrained clustering, used in this study is a powerful tool to analyze urban division. It turns out to be extendible to much more general settings by replacing constraint conditions. Since this method is completely data-driven and does not rely on any prior knowledge regarding urban divisions, it can be considered as an automated information-mining technique, which can help us better understand urban structure under hidden economic/market conditions from data.

(4)    In the aspect of regression analysis, the full-sample OLS result shows that Hangzhou is distinct from Beijing in the relation between subway system and housing price. Weird signs are observed for the coefficients of the metro index and accessibility to metro stations in Hangzhou. Spatial heterogeneity might be a reasonable explanation for that weirdness. By applying constrained clustering, it is found that the housing market in Hangzhou is partitioned into multiple submarkets, and different submarkets have quite different pricing mechanisms. By the comparison between different submarkets, it is found that it is the scale of the metro index that determines in which way metro stations can generate a premium for housing units nearby. Moreover, the spatial variation of the metro index among various submarkets reveals the existence of a deep connection between the hidden features of urban structure and housing market, and the metro index functions as a channel that helps implement the impact of urban structure on housing price.

The current paper also has some limitations: First, although the focus is on the relation between subway system and housing price, it turns out that other public transportation means play a role in influencing that relation; future studies should include the substitutability between subway and bus system. Second, only Beijing and Hangzhou were analyzed in this study; so, the sample might be too restrictive, and a variety of more cities should be included in future studies to account for more versatile structural features that can affect the relationship between subway system and housing market. Finally, incorporating spatio-temporal data into the constrained clustering framework is beneficial [74–76], as the structural change of housing price within every submarket might significantly affect the range and number of submarkets, and keeping track of these changes can help detect the time-varying trend of urban divisions, which gives a more complete description of the dynamic interaction between subway system and urban formation/transformation.

**Author Contributions:** Conceptualization, X.Z. and Y.Z.; methodology, X.Z. and L.S.; software, X.Z. and L.S.; validation, Y.Z. and Q.D.; formal analysis, X.Z. and Y.Z.; investigation, X.Z.; resources, Y.Z. and Q.D.; data curation, X.Z.; writing, original draft preparation, X.Z. and Y.Z.; writing, review and editing, Q.D., L.S., Y.Z., and X.Z.; visualization, Q.D.; supervision, L.S.; project administration, Y.Z.; funding acquisition, Y.Z. and Q.D. The data analyzed in this article are available per request from the first author, through xiaoqizh@buffalo.edu.

**Funding:** This work was partially supported by the National Natural Science Foundation of China under Grant Nos. 71631005 and 71471161 and by the Humanities of Social Science Foundation of Ministry of Education in China under Grant No. 17XJC790001.

**Conflicts of Interest:** The authors declare no conflict of interest.

## Abbreviations

The following abbreviations are used in this manuscript:

GWR    geographically-weighted regression
HPM    hedonic price model

# Appendix A. Tables and Figures

*Appendix A.1. Tables*

**Table A1.** Beijing vs. Hangzhou.

| City | Center Location (Lat,Lon) | GDP (Billion RMB) | Population Size (million) | Built Area (km$^2$) | # County-level Administrative Units | Subway |
|---|---|---|---|---|---|---|
| Beijing | 39.9° N, 116.41° E | 2800 | 21.7 | 1419 | 16 | The first subway line started operating in 1971, and up to November 2017, there are 18 subway lines operating in Beijing. |
| Hangzhou | 30.16° N, 120.12° E | 1255.6 | 9.46 | 541 | 11 | The first subway line started operating in 2011, and up to November 2017, there are 2 subway lines operating in Hangzhou. |

**Table A2.** Summary of the variables.

| Variable | Meaning | Beijing | | | | Hangzhou | | | |
|---|---|---|---|---|---|---|---|---|---|
| | | Min | Max | Mean | Std. | Min | Max | Mean | Std. |
| log unit price | Log of unit price per square meter | 0.12 | 5.17 | 3.4 | 1.28 | −0.2 | 5.12 | 1.97 | 1.18 |
| **Public Transport** | | | | | | | | | |
| log dist subway | Log of distance (km) to the nearest metro station | −2.23 | 3.45 | 0.13 | 1.08 | −2.18 | 3.47 | 0.49 | 1.07 |
| metro lt 1 | Whether there is a metro station within 1 km (1 = yes, 0 = no) | 0 | 1 | 0.58 | 0.49 | 0 | 1 | 0.39 | 0.49 |
| metro lt 2 | Whether there is a metro station between 1 km and 2 km (1 = yes, 0 = no) | 0 | 1 | 0.18 | 0.39 | 0 | 1 | 0.25 | 0.43 |
| log dist bus | Log of the distance (km) to the nearest bus station | −2.29 | 2.9 | −0.89 | 1.12 | −2.26 | 1.79 | −1.37 | 0.58 |
| No. bus routes | Number of bus routes offered by the nearest bus station within 1 km | 0 | 312 | 84.42 | 58.84 | 0 | 400 | 113.01 | 78.65 |
| bus in 1 km | Whether there is a bus station within 1 km (1 = yes, 0 = no) | 0 | 1 | 0.88 | 0.33 | 0 | 1 | 0.97 | 0.16 |
| metro index | The average ratio of traveling time from the nearest metro station to a set of major destinations by non-metro routes to the time vs. by metro-prioritized routes | 0.66 | 2.06 | 1.52 | 0.14 | 0.68 | 1.67 | 1.28 | 0.19 |
| **Construction** | | | | | | | | | |
| log area | Log of construction area (m$^2$) | 2.31 | 4.6 | 3.5 | 0.75 | 2.31 | 4.6 | 3.65 | 0.78 |
| age | The age (years) of the apartment unit (2017 minus the year built) | 0 | 59 | 12.23 | 6.98 | 0 | 47 | 12.18 | 8.69 |
| South | Whether the orientation direction includes south (south, southeast, southwest, etc., 1 = yes, 0 = no) | 0 | 1 | 0.8 | 0.4 | 0 | 1 | 0.96 | 0.21 |

**Table A2.** *Cont.*

| | | Beijing | | | | Hangzhou | | | |
|---|---|---|---|---|---|---|---|---|---|
| **Variable** | **Meaning** | **Min** | **Max** | **Mean** | **Std.** | **Min** | **Max** | **Mean** | **Std.** |
| **Construction** | | | | | | | | | |
| lobby No. | The number of lobby rooms | 0 | 8 | 1.7 | 0.79 | 0 | 5 | 1.66 | 0.56 |
| room No. | The number of bedrooms | 1 | 9 | 2.79 | 1.19 | 1 | 9 | 2.66 | 1.01 |
| log stair | Log of the floor that an apartment is on | −2.3 | 4.04 | 1.35 | 1.41 | −2.3 | 3.8 | 1.65 | 1.03 |
| **Location** | Principal components of log traveling time (seconds) to a set of major destinations | | | | | | | | |
| duration PCA0 | The 1st principal component | −39.95 | −33.85 | −35.68 | 0.96 | −39.17 | −33.98 | −35.55 | 0.85 |
| duration PCA1 | The 2nd principal component | −2.82 | 2.2 | 0.04 | 0.71 | −3.24 | 3.36 | 0 | 0.81 |
| duration PCA2 | The 3rd principal component | −1.71 | 3.01 | 0.01 | 0.59 | −4.32 | 2.84 | 0 | 0.69 |
| duration PCA3 | The 4th principal component | −2.81 | 1.58 | −0.01 | 0.38 | −1.85 | 4.08 | −0.01 | 0.64 |
| duration PCA4 | The 5th principal component | −2.09 | 2.79 | 0 | 0.32 | −2.45 | 2.27 | 0 | 0.55 |
| **Neighborhood** | | | | | | | | | |
| log dist school | Log of the distance (km) to the nearest primary and middle school | −2.3 | 2.92 | −0.37 | 1.04 | −2.3 | 1.79 | −0.63 | 0.68 |
| log dist mall | Log of the distance (km) to the nearest mall | −2.17 | 3.45 | 0.14 | 1.14 | −2.27 | 2.73 | 0.22 | 0.71 |
| log dist hospital | Log of the distance (km) to the nearest hospital | −1.86 | 3.39 | 0.89 | 0.83 | −2.09 | 2.88 | 1.1 | 0.79 |

**Table A3.** Full sample results for Beijing and Hangzhou.

| | Beijing | Hangzhou |
|---|---|---|
| **Variable** | **Coef.** | **Coef.** |
| intercept | 5.471 *** | 13.611 *** |
| area | −1.04 *** | −1.088 *** |
| age | −0.009 *** | −0.002 *** |
| stair | −0.005 | −0.006 |
| orientation | 0.034 | 0.04 |
| lobby No. | 0.279 *** | 0.243 *** |
| room No. | 0.375 *** | 0.352 *** |
| dist bus | −0.061 ** | 0.105 *** |
| bus in 1 km | 0.163 | 0.341 *** |
| No. bus routes | −0.06 *** | 0.009 |
| duration PCA0 | 0.005 | 0.253 *** |
| duration PCA1 | −0.114 *** | −0.06 *** |
| duration PCA2 | 0.271 *** | 0.137 *** |
| duration PCA3 | −0.56 *** | 0.001 |
| duration PCA4 | 0.105 *** | −0.005 |
| dist school | −0.071 *** | −0.015 |
| dist mall | −0.132 *** | 0.017 |
| dist hospital | 0.0286 | 0.038 *** |
| dist subway | 0.056 ** | −0.069 *** |
| metro lt 1 | 0.105 * | −0.082 ** |
| metro lt 2 | −0.045 | −0.046 * |
| metro index | 0.159 ** | −0.17 *** |
| Adj. $R^2$ | 0.852 | 0.886 |
| *F*-statistic | 649.5 *** | 1573 *** |
| Obs. | 2359 | 4130 |

*: 10% significant, **: 5% significant, ***: 1% significant.

**Table A4.** Regression results by cluster (Hangzhou).

| | Satellite Cluster | Core Cluster |
|---|---|---|
| **Variable** | **Coef.** | **Coef.** |
| intercept | 14.507 *** | 10.086 *** |
| area | −1.079 *** | −1.086 *** |
| age | 0.003 ** | −0.008 *** |
| stair | 0.03 *** | −0.027 *** |
| orientation | 0.1375 | −0.032 |
| lobby No. | 0.258 *** | 0.226 *** |
| room No. | 0.35 *** | 0.354 *** |
| dist bus | 0.09 *** | 0.079 *** |
| bus in 1 km | 0.722 *** | 1.179 |
| No. bus routes | −0.109 *** | −0.006 |
| duration PCA0 | 0.296 *** | 0.179 *** |
| duration PCA1 | 0.102 *** | −0.082 *** |
| duration PCA2 | 0.0521 *** | 0.119 *** |
| duration PCA3 | −0.265 *** | −0.034 * |
| duration PCA4 | −0.255 *** | 0.075 *** |
| dist school | −0.123 *** | 0.017 |
| dist mall | 0.079 *** | −0.011 |
| dist hospital | −0.077 *** | 0.026 * |
| dist subway | −0.033 | −0.175 *** |
| metro lt 1 | −0.017 * | −0.099 ** |
| metro lt 2 | 0.023 * | −0.06 * |
| metro index | 0.207 *** | 0.185 ** |
| Adj. $R^2$ | 0.901 | 0.896 |
| *F*-statistic | 734.1 *** | 997 *** |
| Obs. | 1699 | 2431 |

*: 10% significant, **: 5% significant, ***: 1% significant.

*Appendix A.2. Figures*

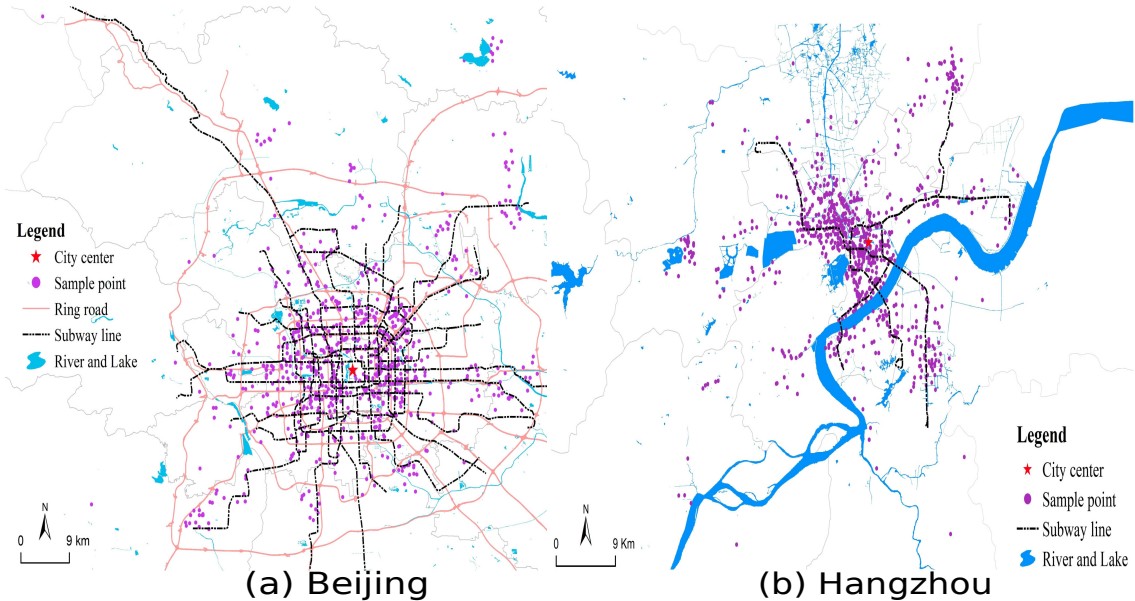

**Figure A1.** Study area and spatial distribution of samples.

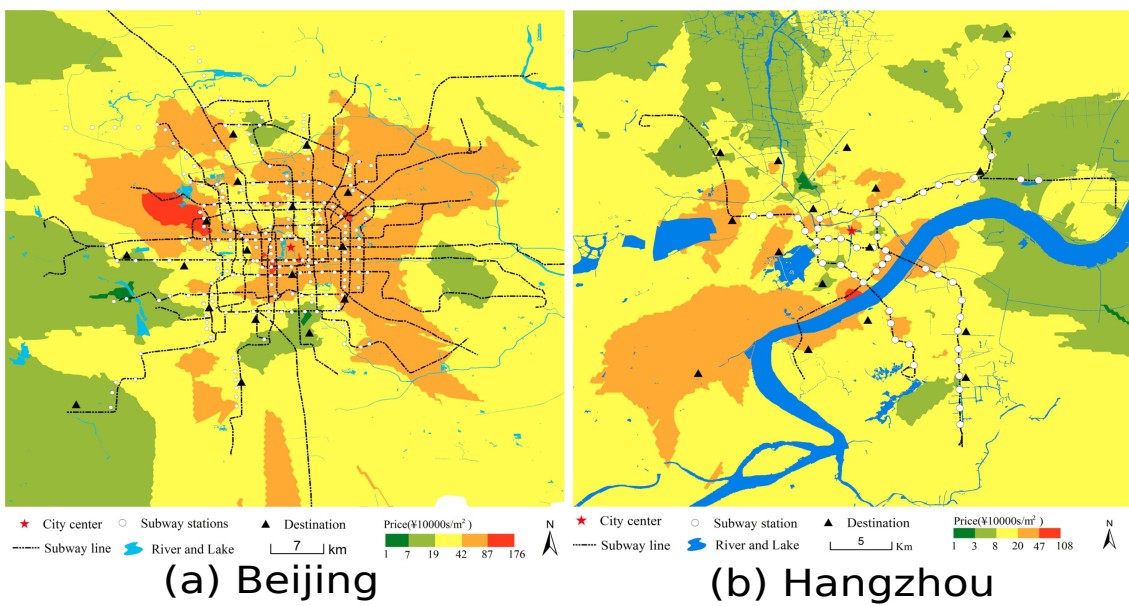

**Figure A2.** Spatial distribution of housing unit price (yuan/m$^2$) and metro stations in Beijing and Hangzhou.

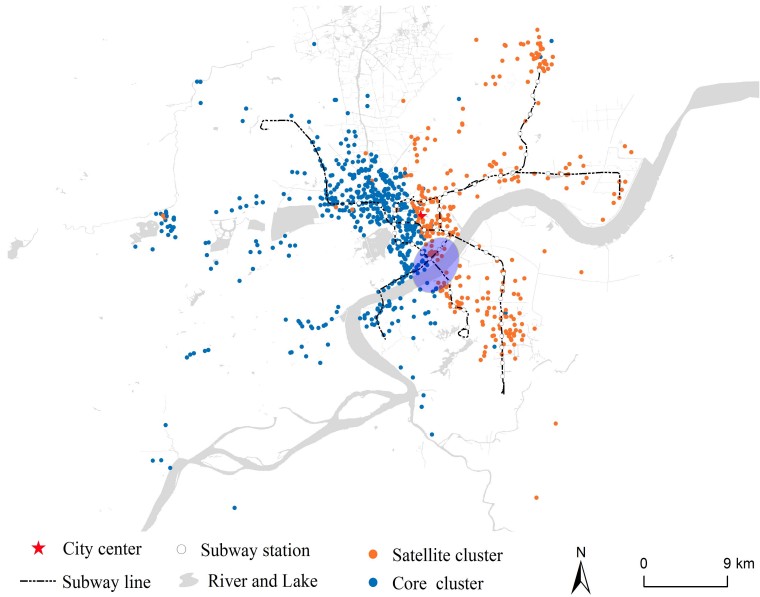

**Figure A3.** Clustered regions of Hangzhou.

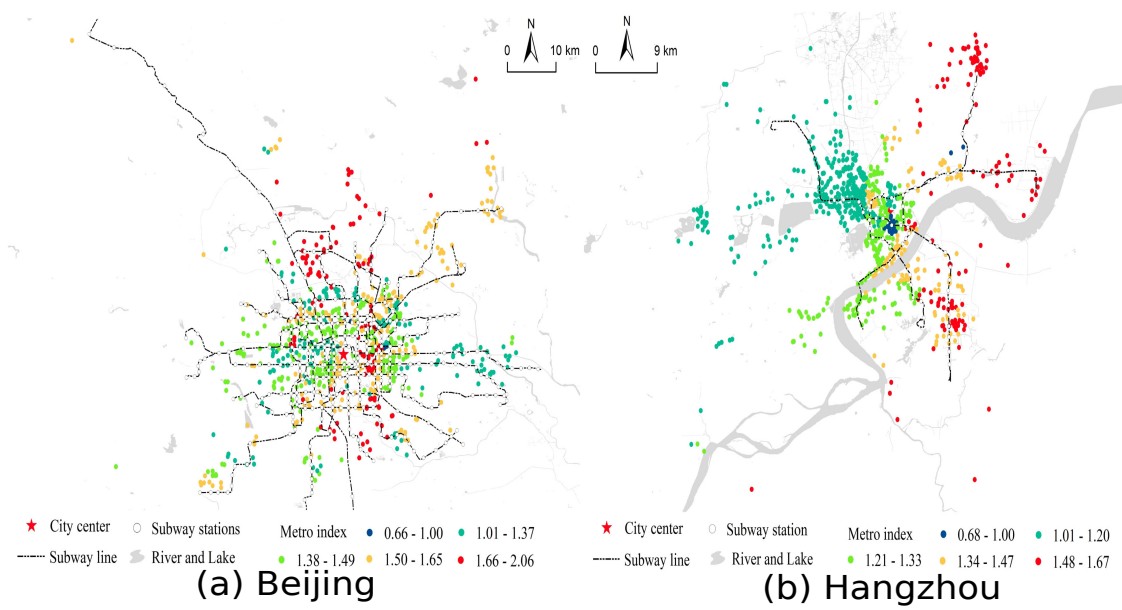

**Figure A4.** Spatial distribution of metro index in Beijing and Hangzhou.

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
