# Peer review of "Urban Structure, Subway Systemand Housing Price: Evidence from Beijing and Hangzhou, China"

_sustainability, doi:10.3390/su11030669_

Round 1

Reviewer 1 Report

Please see my report.

Author Response

Thank you for the comments on our manuscript entitled “Urban structure, subway system and housing price: evidence from Beijing and Hangzhou, China” (ID: sustainability-423215). Those comments are very helpful for improving our paper. We have studied the comments carefully and made revisions accordingly. Below is the response to the reviewers’ comments.

Comment (1) from Reviewer:

More recent research, such as Huang et al. (2018). Leung et al. (2014), all found the subway station to be important for housing prices. Besides, it is virtually impossible to distinguish the effect of subway station from railway station because in some areas, they are geographically close. The revised edition should correct that station and relate to these papers.

Response from Authors:

The reference paper has been added.

We didn’t mention anything related to railway station, the term “rail transport” can also refer to the transportation by subway in literature, to eliminate misunderstanding, we have replace the word “rail transport” by “subway transport”.

Comment (2) from the Reviewer:

In fact, Tse and Chan (2003) measures the commute time instead of the commute distance and how it would affect the property price gradient. And the latter is clearly related to the urban structure, especially in the context of mono-centric city. The revised edition of the paper should relate to that paper.

Response from the authors:

Reference and comments to the mentioned paper has been added.

Comment (3) from Reviewer:

It is not clear whether those are (a) listing prices only, (b) transaction prices only, or (c) a combination of listing and transaction prices. In the literature, it is well known that the listing price can be a strategy used by the sellers, while the transaction price is the part of the equilibrium. For instance, Leung et al. study the ratio of the listing and transaction price and relate it to the time-on-the-market (TOM), i.e. the time between the property is listed and the time the property is actually transacted and find that there is a clear statistical relationship. The revised edition should clarify this point and relate to the literature.

Response from authors:

As we have highlighted in the manuscript that information regarding the listed apartment is extracted, so the price is the listed price. Unlike the reference paper mentioned by the reviewer, the current paper focuses completely on the spatial analysis, instead of the spatio-temporal analysis. So only the price on a cross section associated a fixed time point is selected. For our data, we only extracted the listed apartments within the last week of Oct. 2017 from fang.com, because we don’t want to include the price fluctuation that is usual in Chinese housing market into our spatial analysis, while the time range chosen to be one week instead of one day is to guarantee that there are sufficient amount of records to cover the entire urban area of both cities. The  selection of the listing price can be justified as the following: (1) there are not sufficient amount of transaction records within a short time period (cross section), and the geographic coverage of transaction records within a relative short time range is not even, i.e. the distribution of transactions is sparse in some areas of the cities but dense in the other, which might cause bias; (2) Although the listing price might be strategically used by sellers and becomes systematically higher than the real price, but it is also known that the transaction price can be strategically used by buyers for the purpose of reducing tax burden and is usually much lower that the real price; on the other hand, no matter which price is used, as long as the observed price is systematically higher or lower than the true price, it won’t affect the city-level spatial analysis. So based on (1) and (2), we think it is not quite problematic to concentrate on the listing price.

Finally, this paper is design to reveal the spatial relation between housing price and subway stations/subway system, and how the features of urban structure can influence that relationship. The concept of time has nothing to do with the desired spatial relationship, as long as it only affects the level but does not significantly affect the shape of the spatial distribution of price within every submarket. Of course, for a fast-developing city, such as Hangzhou, all of the spatial relation, the impact of urban structure and even the urban structure itself can change over time. So it is definitely meaningful as mentioned by the reviewer to include the price at different time points and re-do the spatial analysis and compare the temporal variation of the hedonic coefficients and clustering results, which has already been included in a new paper that we are currently working on. But for the current paper, we think it is better to keep focus on the spatial aspect of the topic, instead of deal with both the spatial and temporal aspects.

Comment (4) from reviewers:

Another issue is about the identification of the “hot spot.” The paper writes .....

The paper should clarify more. For instance, are those data also collected only in Oct., 2017? Or in a different time period? And it is not clear what it means for “more intense clustering.” For instance, some spot in Beijing and Hangzhou may have a lot of tourists in some seasons, but not other seasons. Some districts may have a lot of people working there, but may be almost empty during, say, Lunar New Year.

Some districts may have a lot of high apartment buildings for residents, but not much commercial activities, except for grocery shops. And some major transit point such as the airport or major train station would have a lot of people come and go all day long, but not much economic activities there, except for meals and selling gifts. How should we compare these different types of places? The paper should clarify more.

Response from authors:

There is a misleading, the “hot-spot analysis” is one word which stands for a well-known technique in geographic analysis to detect in which location a kind of spatial feature is more densely clustered. n this paper, the feature is selected as the density of point of interests (POI). The reviewer interprets the “hot-spot” as the tourist spot, which is not correct. Throughout the entire paper we didn’t deal with the tourist spots or landscapes, which can definitely be discussed for more details in a separated paper.

The “hot-spot analysis” together with the choice of POIs can definitely help resolve the issue of “economic activities” as pointed out by the reviewer, which has also already been contained in the design of metro index and commuting time PCA variables (please check the section 3.3.1). In fact, as we have described as in the manuscript, the locations and types of POIs are selected from Baidu place API, where the types of POIs are chosen to be shopping malls, hospitals, schools, restaurants and so on, which are exactly those places where economic activities happen. Using the locations of POIs, we can derive the spatial density of POIs by the kernel smooth method as we have introduced in the equation (2). Finally, taking the density as the spatial feature, applying the hot-spot analysis, we pinpointed a set of locations that have high density of POIs and meanwhile are enclosed by locations that also have high density of POIs (this is the real meaning of the term “hot-spot” within the context of hot-spot analysis), where can be considered as the regions that have the most economic activities in Beijing and Hangzhou.

Comment (5) from reviewer:

I understand that the authors may want a fair comparison and hence choose 18 major destinations in both cities. On the other hand, the two cities are very different. It is clear from Table A1 that even restricted to “core area,” Beijing is almost three times as Hangzhou. Hence, is it a sensible to choose the same amount of major destinations for the two cities.

Also clear from Table A1 is that in Beijing there are 18 subway lines and Hangzhou only has 2. While the population of Beijing is bigger and has more population (about twice) than Hangzhou, the density of subway stations of Beijing should be much higher than Hangzhou. 

Also, the location of that lines is not random. The government might have certain considerations before they build the subway lines. It may be especially important for Hangzhou, where the whole city has only 2 lines. (In comparison, Guangzhou has slightly more population and probably has 10 subway lines).

Response from authors:

Based on the locations selected from hot-spot analysis, we applied a K-means clustering with K=18 to identify 18 major destinations and used them to construct the commuting time variables and the metro index. As pointed out by the reviewer, due to the difference between Beijing and Hangzhou, selecting K=18 might be problematic. But the choice can be justified as the following: (1) as we mentioned in the paper, “based on preliminary experiment, 18 is the largest number that can guarantee all destinations not too close to each other in both Beijing and Hangzhou”; if two locations are too close to each other, it will cause multi-colinarity issue for regression, while if two locations are too far away from each other, there might be loss of information; so the choice of 18 is driven by the data instead of keeping fair for both cities; (2) the principle component analysis (PCA) is applied to extract five commuting time PCA variables to further reduce the potential multi-colinarity; Similarly the number 5 is also selected in a data-driven manner as discussed in the paper that the first 5 principal components can explain 95% total variance of commuting time to all the 18 destinations. So, the final choice of major destinations and the commuting time PCA variables is completely driven by the data, consequently the difference of the two cities should have already been encoded into the construction of these variables.

Of course, we cannot guarantee that all potential difference between Beijing and Hangzhou have been accounted for, it would be helpful in a new study that include more city-specific variables to cover the desired difference.

For the endogeneity issue of locations of subway stations, we agree that there exists endogeneity, the government may construct subway stations in suburb or exburb area to lift up the local housing price. This is particularly true for Hangzhou as the subway line is designed to connect the urban core with its new satellite cities,  which forms a main reason to apply the constrained clustering technique to partition the urban area

In fact, the endogeneity of subway station can be interpreted through the way that in different places the pricing mechanism for apartments is different, i.e. price in the low price area (suburb area) may depend on different factors in different way compared to that in the high price place (core region). In the other words, the endogeneity issue of subway station is reflected as the heterogeneity of different housing submarkets. The constrained clustering technique can help effectively distinguish heterogeneous housing submarkets, and reveal the distinct pricing mechanism. For instance, the clustering result in Hangzhou implies that the housing submarket coinciding with the old urban core has the price negatively response to being close to subway station, in contrast, in the submarket corresponding to the new cities in east and southern east part of Hangzhou, price reacts positively. This distinction, by and large, has already captured the endogeneity of subway station mentioned by the reviewer, if we focus merely within every submarket and do the regression (as we have done in the paper), there won’t be any concern of endogeneity.

In addition, if we look more carefully at the clustering result, it can be seen that the result conveys more subtle information of housing submarket segregation which is beyond simply comparing the high and low price. Apartments within the area (marked by the shadowed circle in Figure A3) has very high unit price, but this area is still assigned to the satellite cluster. This is because this area is the new CBD of Hangzhou which is still under-developed by now. Residents living close to this area are not really working there, then strong commuting demand is arisen which makes home-buyers in this area behave more analogous to home-buyers in the satelite cities rather than those in the core region, which finally determines the membership of the housing submarket of this area. So, we believe the constrained clustering proposed in this paper can better resolve the endogeneity issue than some alternative econometric methods. The similar discussion has already been included in the section 4.3.2 of the manuscript.

Comment (6) from Reviewer:

The paper is correct to point out the submarket issue and proposes a procedure to study it. It should be noted that the relationship among different sub-markets can vary over the housing market cycles (i.e. when the market goes up and down). Leung et al. (2013) provide a case‐study of Hong Kong, which is based on more than 200,000 transactions. The paper should qualify their results and explain how sensitive their procedure could be affected by the housing cycle and relate to the literature.

Also, due to its geographical or residential composition, some sub‐markets may be more frequent to have fire-sale (Leung and Zhang, 2011) or speculation (Leung and Tse, 2017). In that case, the price effect may be over or under‐estimated. Taking data of one month only is dangerous. The authors should either expand their dataset, or recognize these limitations and qualify their results in the paper.

Response from authors:

Similar to the response to comment (3), the paper focus completely on the spatial relation between housing price, subway and urban structure, so the clustering is conducted on the basis of a cross section data set. So the temporal change of housing price is not studied deliberately. Also as mentioned in the response to comment (3), if the housing market cycle only affects the overall price level within each submarket, but does not change the relative price between different locations within the same submarket (i.e. only affect the level of the spatial distribution function of price, but no impact on the shape of it), then neither the clustering result of submarkets nor the regression coefficients within the hedonic model will be changed. So the result is robust with respect to price change at the submarket level.

But in reality, it can be expected that there are many causes, such as the new urban planning policy, that can make the price of different locations within the same submarket change differently over time, this kind of price change is called structural change. Structural change can definitely change both of the range and number of submarkets as well as the regression coefficients within each submarket. But we think it is more proper to include the structural change into a new paper, because the goal of the current paper is to detect the spatial relationship among housing price, subway and urban structure, instead of how price reacts to the change of urban planning policy and/or the change of urban structure.

Finally, some discussions are added in the end of the paper to discuss the limitation and potential extensions of the current paper, the literature pointed out by the review is also added there.

Reviewer 2 Report

I think this is an interesting paper with a number of new features. I have only a few concerns that the authors should comment on:

- The effect of subway stations can be interpreted in several ways: e,g, the effect through shorter commuting times or  an effect given a certain commuting time (because of transport mode preferences). How these are separated should be commented upon. The metro index is an interesting concept but it seems problematic if it is included together with commuting times. Some clarification is needed. 

- The effect of subway stations should be related to the overall transportation policy of the city. Are e.g other areas compensated with good bus transport there should not be any reason to expect a major effect. Some comments on the official transportation policy in the cities would be welcome.

- The location of a subway station may be endogenous in relation to house prices as building a subway may be a way of lifting areas with low house prices (low for other reasons that commuting times). If the effect is not large enough house prices may remain rather low even if they are higher than otherwise. Some comments about this would also be welcome.

Author Response

Thank you for the comments on our manuscript entitled “Urban structure, subway system and housing price: evidence from Beijing and Hangzhou, China” (ID: sustainability-423215). Those comments are very helpful for improving our paper. We have studied the comments carefully and made revisions accordingly. Below is the response to the reviewers’ comments.

Comments from reviewer:

I think this is an interesting paper with a number of new features. I have only a few concerns that the authors should comment on:

- The effect of subway stations can be interpreted in several ways: e,g, the effect through shorter commuting times or  an effect given a certain commuting time (because of transport mode preferences). How these are separated should be commented upon. The metro index is an interesting concept but it seems problematic if it is included together with commuting times. Some clarification is needed. 

Response from authors:

The effect of subway station in this manuscript is referring to provide "shorter commuting time", with which the metro index is designed to catch up. As for the transport mode preference, it is not likely to distinguish the group of passengers that prefer a certain type of commuting tool than the others. But in both Hangzhou and Beijing, subway and bus are two most popular transportation tools and there is not a huge gap between the cost to take subway and the cost to take bus, so it can be expected that the commuting efficiency of subway comes mainly from the shorter commuting time whenever it is possible. 

For the issue of using commuting time to main destinations together with metro index, my justification is as the following: (1) metro index is the mean of a set of ratios of two commuting times; for each of the ratios, the denominator is the commuting time by subway route to a main destination which is the same as used in the construction of the commuting time PCA variables; while the numerator is the the commuting time to the same destination by the route that subway is excluded, which could be and actually is significantly different from the denominator, this difference makes the metro index largely independent from the commuting time PCA variables. (2) We have already applied the principal component decomposition (PCA) method to remove the dependence among all commuting times variables, so the selected commuting PCA variables shouldnt have strong correlation with metro index, so it is not problematic to include both of them in regression. (3) The commuting PCA variables is nothing more than a distance measure between the given apartment and a destination by which we can describe the location attribute of the apartment. We can definitely replace the commuting time PCA variables with the Euclidean distance PCA variables which seems to be less inter-dependent with the metro index. In fact, we have done so in preliminary analysis, but the fitting by Euclidean distance model is less accurate than the commuting time model, which is as expected for the meta-city such as Beijing and Hangzhou, so in the final version we consider commuting time model only.

Comments from the reviewer:

- The effect of subway stations should be related to the overall transportation policy of the city. Are e.g other areas compensated with good bus transport there should not be any reason to expect a major effect. Some comments on the official transportation policy in the cities would be welcome.

Response from the authors:

The total number of bus route near to every apartment and whether there exists a bus stop in the area within 1km from every apartment are already included in the regression in order to control the effect of the alternative transport mode, the result has been reported in the table A3 and A4, so the conclusion is drawn on the basis that bus has been included. 

Except for bus and subway, the only popular alternative transport mode is taxi, but the use of taxi does not quite depend on the location. Because the study area in both Beijing and Hangzhou are either within or close to the urban core region of the two biggest cities, Beijing and Hangzhou, in China, it is reasonable to expect that places within the entire study area have almost the same accessibility to taxi, so there won't be a significant change for the results even if the taxi is included in the regression. 

Comments from reviewer:

- The location of a subway station may be endogenous in relation to house prices as building a subway may be a way of lifting areas with low house prices (low for other reasons that commuting times). If the effect is not large enough house prices may remain rather low even if they are higher than otherwise. Some comments about this would also be welcome.

Response from authors:

Yes, the location of a subway station is highly likely to be endogenous, this is particularly true for Hangzhou as the subway line is designed to connect the urban core with its new satellite cities,  which forms a main reason to apply the constrained clustering technique to partition the urban area

In fact, the endogeneity of subway station can be interpreted through the way that in different places the pricing mechanism for apartments is different, i.e. price in the low price area (suburb area) may depend on different factors in different way compared to that in the high price place (core region). In the other words, the endogeneity issue of subway station is reflected as the heterogeneity of different housing submarkets. The constrained clustering technique can help effectively distinguish heterogeneous housing submarkets, and reveal the distinct pricing mechanism. For instance, the clustering result in Hangzhou implies that the housing submarket coinciding with the old urban core has the price negatively response to being close to subway station, in contrast, in the submarket corresponding to the new cities in east and southern east part of Hangzhou, price reacts positively. This distinction, by and large, has already captured the endogeneity of subway station mentioned by the reviewer, if we focus merely within every submarket and do the regression (as we have done in the paper), there wont be any concern of endogeneity.

In addition, if we look more carefully at the clustering result, it can be seen that the result conveys more subtle information of housing submarket segregation which is beyond simply comparing the high and low price. Apartments within the area (marked by the shadowed circle in Figure A3) has very high unit price, but this area is still assigned to the satellite cluster. This is because this area is the new CBD of Hangzhou which is still under-developed by now. Residents living close to this area are not really working there, then strong commuting demand is arisen which makes home-buyers in this area behave more analogous to home-buyers in the satelite cities rather than those in the core region, which finally determines the membership of the housing submarket of this area. So, we believe the constrained clustering proposed in this paper can better resolve the endogeneity issue than some alternative econometric methods. The similar discussion has already been included in the section 4.3.2 of the manuscript.

Round 2

Reviewer 1 Report

Please see the report.

Author Response

Thank you for the comments on our manuscript entitled “Urban structure, subway syste, and housing price: evidence from Beijing and Hangzhou, China” (ID: sustainability-423215). Those comments are very helpful for improving our paper. We have studied the comments carefully and made revision accordingly. Below is the response to the reviews’ comments.

Comments from reviewer:

My concern is, if we repeat the exercise in other months, i.e. months other than October, are we going to get the same set of “point of interest” (POI). Statistically, the only way to find out is to indeed repeat the search of POI in all other months and then tabulate them. We can them see whether the POI identified in different months are exactly the same. Notice that there are research on “seasonal cycles.” The economic activities generated in different months can be very different, even without location change. In some Western countries, for instance, the “Christmas effect” can be so large that the rental in certain months are multiples of the other months. The same can also happen in China. Therefore, the authors should check whether the identified POI are robust.

Response from authors:

As mentioned by the reviewer that POIs may change over time, but in general, this change does not really matter, which can be verified by the following:

(1)  The types of POIs are selected to be shopping malls, schools, hospitals, restaurants or the like, which can definitely change over time (for instance, a shopping mall is replaced with a school), but changes of this kind occur very rarely, they even cannot occur over years. Hence, the POI data can be thought of as fixed when we study housing market.

(2)  Even if changes of POIs can happen within a couple of months, it is still not possible to affect the result of the paper. Because what really matters is not whether or not a shopping mall at a location is replaced with another type of POI. It is the spatial distribution density of POIs that matters, as described in section 3.3.1. So, as long as changes of POI locations do not affect the overall distributional density, they won’t affect our result. For the change of density of POIs, it is almost impossible to happen within tens of years, this robustness makes POI data widely used in geographic literature as the background information.

In sum, we believe it is not necessary to worry about the changes of POIs.

In addition, as we highlighted in the round-1 response, the paper concentrated on the spatial relationship among urban structure, subway system and housing price, the housing market cycles are not a major concern within such a spatial analytic setting. We also verified that when housing market cycles only impact the price level but do not significantly change the relative housing prices within every submarket, the analysis (including both the partition of submarkets and the regression within every submarket) would still be valid. So, we believe in the context of the current paper, it is not proper nor necessary to include the temporal effect of housing market cycles. But the time series data of housing prices can definitely be used in another independent paper, in order to answer some other fundamental questions in housing studies, such as how the spillover of housing price occurs within/among submarkets, which we will work on in future.